# Use of Intravaginal Progesterone-Releasing Device Results in Similar Pregnancy Rates and Losses to Long-Acting Progesterone to Synchronize Acyclic Embryo Recipient Mares

**DOI:** 10.3390/vetsci8090190

**Published:** 2021-09-10

**Authors:** Lorenzo G. T. M. Segabinazzi, Luiz R. P. Andrade, Marco A. Alvarenga, Jose A. Dell’Aqua, Igor F. Canisso

**Affiliations:** 1Department of Veterinary Surgery and Animal Reproduction, School of Veterinary Medicine and Animal Science, São Paulo State University, UNESP, Botucatu 18618-681, Brazil; lgseg@hotmail.com (L.G.T.M.S.); juniorandrade53@hotmail.com (L.R.P.A.J.); marco.alvarenga@unesp.br (M.A.A.); dellaquajunior@uol.com.br (J.A.D.J.); 2Department of Veterinary Clinical Medicine, College of Veterinary Medicine, University of Illinois, 1008 W Hazelwood Drive, Urbana, IL 61802, USA

**Keywords:** equine, recipient mare, embryo transfer, fertility, hormonal therapy

## Abstract

The objectives of this study were: (1) to assess uterine features and serum progesterone concentrations of acyclic mares synchronized and resynchronized with intravaginal progesterone release device (IPRD), and (2) to compare pregnancy rates and losses of cyclic and acyclic embryo recipient mares treated with different synchronization protocols. In Experiment 1, mares (n = 12) received estradiol for 3 days (E2-3d), and then 24 h after the last injection, an IPRD was inserted and kept in place for 9 days. Three days after IPRD removal, mares were treated with E2-3d, and then a new IPRD was inserted and maintained for three days. Serum progesterone concentrations were assessed 2, 6, and 12 h after insertion and removal of IPRD, and then daily from the insertion of the first IPRD to one day after removal of the second IPRD. Experiment 2 was conducted with embryo recipient mares randomly assigned to four groups: (1) Cyclic: mares (n = 75) had ovulation confirmed after receiving a single dose of histrelin when a periovulatory follicle was first detected, (2) LAP4: acyclic mares (n = 92) were treated with E2-3d and then administered a single dose of LAP4 24 h after the last estradiol injection, (3) IPRD: acyclic mares (n = 130) were treated with E2-3d and an IPRD for 4–8 days, and (4) RE-IPRD: acyclic mares (n = 32) were synchronized as in the IPRD group but not used for embryo transfer (ET), then 8 to 15 days later, the mares were resynchronized with E2-3d and an IPRD for 4–8 days. In vivo-produced Day-8 embryos were collected and transferred 4–8 days after ovulation or progesterone treatments. Mares in IPRD and RE-IPRD groups had the intravaginal device removed immediately before ET, and then a new IPRD was inserted right after ET. Pregnancy diagnosis was performed at 5, 30, and 60 days after ET. Once pregnancy was confirmed, mares in the three acyclic groups received weekly injections of LAP4 (1.5 g) until 120 days of pregnancy. Mares in IPRD and RE-IPRD groups had the device removed three days after the first pregnancy diagnosis. In Experiment 1, progesterone concentrations increased rapidly starting 2 h after insertion of IPRD (*p* < 0.05); then, concentrations plateaued well above pregnancy maintenance until removal on days 9 and 3, respectively. Progesterone concentrations were reduced to baseline 24 h after IPRD removal (*p* < 0.05). For experiment 2, there was no difference in pregnancy rates across groups (65–74%) or pregnancy losses by 60 days of gestation (7–12%) (*p* > 0.05). In conclusion, the IPRD used herein resulted in a rapid increase and a sharp decline in progesterone concentrations upon its insertion and removal, respectively. Finally, our results demonstrated that IPRD could be a compatible alternative to LAP4 to synchronize and resynchronize acyclic embryo recipient mares.

## 1. Introduction

Embryo transfer (ET) is an assisted reproductive technique used to maximize the number of offspring from mares with a desired phenotype. Embryo donor mares can also be enrolled in ET programs due to health concerns (i.e., inability to carry a pregnancy to term), or while still competing or showing [1,2,3,4]. Whilst the management and fertility of embryo donor mares, semen quality, transfer technique, and practitioner’s experience can affect the success of an ET program, the management and quality of embryo recipient mares play a major role in the success of ET [1,4,5].

Synchronization between embryo donor and recipient mares is essential for ET programs [4]. Satisfactory pregnancy rates can be achieved using recipient mares ovulating from 4 to 9 days before ET [5,6,7]. Mares are classified as long-day breeders and seasonal polyestrous, and as many as 80% of mares display a seasonal anestrus from October to March and April to September in the northern and southern hemispheres, respectively [8]. In the man-imposed breeding season, January 1 and August 1 are officially recognized as the day-of-birth for foals in the Northern and Southern hemispheres, respectively. Typically, early in the man-imposed breeding season, embryo donor mares are displaying regular ovarian cyclic activity as they have been well-fed during the winter and kept inside stalls under lights, whereas since embryo recipients are kept outside and marginally fed under natural climate, they are not cycling regularly early in the breeding season [5]. Previously, physiological anestrus was a limitation in using acyclic mares as embryo recipient mares. However, studies demonstrated that acyclic mares could be successfully used as embryo recipient mares as long as some type of progestin is administered 4–7 days before and continuously for 70–120 days after ET [6,9,10,11]. In addition, studies also demonstrated that priming acyclic mares with estrogen before progestins can be beneficial for ET results [12,13,14,15].

Long-acting progesterone (LAP4) is the most commonly used hormone to synchronize acyclic recipient mares for ET [16]. Commercially available L4P4 products consist of progesterone conjugated with proprietary molecules, delaying its metabolization [17]. While using LAP4 is convenient, anecdotal clinical experiences suggest that acyclic mares synchronized with LAP4 not used for ET for various reasons (e.g., donor yielding a negative embryo flush or nonviable embryo) do not respond to resynchronization shortly after. Mares receiving LAP4 may not display endometrial edema for days following the last injection, presumably due to the residual LAP4 in plasma. The inability to resynchronize recipients if not transferred or pregnant results in major economic losses to commercial ET programs. Therefore, alternatives are warranted to circumvent this issue.

Intravaginal progesterone-releasing devices (IPRD) are commercially available products labeled for use in cattle [18]. The IPRD are typically made with a silicon matrix loaded with non-synthetic, free (non-bound) progesterone. In horses, IPRD have been used to hasten cyclicity in transitional mares, and to inhibit estrus in cyclic mares [19,20,21,22,23]. Once inserted, IPRD devices rapidly increase progesterone concentration in plasma [22,24,25]. Plasma concentrations of progesterone drop rapidly after removal of the device [22,23]. Therefore, we hypothesized that IPRD can be used to synchronize and resynchronize acyclic recipient mares for ET. The objectives of this study were: (1) to assess uterine features and serum progesterone concentrations of acyclic mares synchronized and resynchronized with an intravaginal progesterone release device (IPRD), and (2) to compare pregnancy rates and losses of cyclic and acyclic embryo recipient mares treated with different synchronization protocols.

## 2. Materials and Methods

All experimental procedures performed in this study were revised and approved by the Animal Care and Use Committee of São Paulo State University. Experiment 1 was conducted at the Department of Veterinary Surgery and Animal Reproduction of São Paulo State University in July 2017. Experiment 2 was conducted in a private embryo recipient farm from 2016 to 2019. The owner signed a consent form permitting us to use the animals in this experiment. 

### 2.1. Experiment 1

#### 2.1.1. Animals and Husbandry

Twelve crossbreds of light breeds, seasonal anestrus mares (8 ± 3 years-old), were enrolled in the experiment. Seasonal anestrus was defined as follicles <20 mm of diameter, serum progesterone concentration <1 ng/mL, and absence of endometrial edema and corpus luteum [24,25] after two consecutive evaluations one week apart. All mares enrolled in the study had a satisfactory body condition score (±7) [26] and belonged to the Department of Veterinary Surgery and Animal Reproduction of São Paulo State University (latitude 22°53′09″ S and longitude 48°26′42″ W). Mares were kept on tropical pastures, received 2.0 kg of commercially available grain, and had free access to water and trace minerals.

#### 2.1.2. Design

Mares received estradiol-17β (17 βeta^®^, Botupharma, Botucatu, Brazil), for three consecutive days (Day –3 [D–3], Day –2 [D–2], Day –1 [D–1]), and then 24 h after the last injection, a commercially available IPRD (Sincrogest, Ouro Fino, Brazil) was manually inserted into the vagina (Day 0 [D0]) and kept in place for 9 days (D9) (Figure 1). The IPRD consisted of a silicone matrix containing 1 g of natural progesterone. Immediately before insertion, each IPRD had the string removed and was sprayed with oxytetracycline and hydrocortisone (Terra-Cortril Spray, Zoetis, Brazil) to mitigate vaginal inflammation, as previously described by Polasek and collaborators [27]. Immediately before removal of the IPRD, the vulva was scrubbed, opened, and the caudal vagina was inspected for hyperemia. Then, one of the authors (L.G.T.M.S.) inserted his hand vaginally and scooped out the IPRD and any possible vaginal secretions. While having the IPRD and secretions in hand, the same author scored the vaginal inflammation based on an arbitrary scale developed for the study. The score was as follows: Absent—no inflammation, as no vaginal secretions could be scooped, or irritation of the caudal vagina seen, Mild—hyperemia of the caudal vaginal mucosa with no discharge, Moderate—hyperemia of the caudal vagina and cloudy discharge, and Severe—hyperemia of the caudal vagina and mucopurulent discharge covering the device.

Three days after removal of the IPRD, mares were resynchronized with three consecutive days of estradiol-17β, as described above, and a new IPRD was vaginally inserted and kept in place for three days (Figure 2). The IPRD was kept in place for three days to simulate a short resynchronization. Transrectal palpation and ultrasonographic evaluations were performed daily from the first day of estrogen administration (D−3) until removal of the second IPRD (Day 18 [D18]). Uterine tone and endometrial edema were scored during each transrectal palpation and ultrasonographic examination respectively, as previously described [7]. Briefly, no visible endometrial edema was deemed as score 0, and exacerbated endometrial edema characterized by a marked orange-slice appearance with rounded edged endometrial folds was deemed as score 4. Intermediate scores had variations in the orange-slice appearance. Uterine tone score 1 had a flaccid and saggy uterus (anestrus), whereas score 3 had a well-toned rounded and tubular uterus. Blood samples were collected 2, 6, and 12 h after insertion (D0 and D15) and removal (D9 and D18) of IPRD, and also daily for 16 consecutive days (D0 to D12, and D14 to D18) (Figure 2).

#### 2.1.3. Progesterone Evaluation

Blood samples were collected via puncture of the external jugular vein for determination of progesterone concentrations. After collection, blood samples were allowed to clot at room temperature for 1 h and then centrifuged at 2000× *g* for 10 min, and then serum was stored at −20 °C until evaluation. Progesterone concentrations were determined with a commercially available radioimmunoassay (RIA Progesterone, Beckman Coulter Company, Chaska, MN, USA) following the manufacturer′s recommendations. The intra-assay coefficient of variation was 2.2%.

### 2.2. Experiment 2

This experiment was conducted in a privately owned ET facility in Pedregulhos, São Paulo, Brazil (latitude 20°15′25″ S and longitude 47°28′36″ W), during three consecutive breeding seasons of the southern hemisphere (August to March). Crossbreds of light-breed embryo recipient mares, 3- to 12-years-old, were kept on pasture of Tifton 85 with free access to water and trace minerals. Mares were randomly assigned to four groups: (1) Cyclic: mares (n = 75) having ovulation confirmed after induction of ovulation with histrelin acetate (250 µg, IM, Strelin, Botupharma, Botucatu, Brazil) when a periovulatory follicle (≥35 mm) was detected, (2) L4P4: acyclic mares (n = 92) were treated with estradiol-17β for three days (as above) and then administered a single dose of LAP4 (P4-300, 1.5 g, IM, Botupharma) 24 h after the last estradiol injection, (3) IPRD: acyclic mares (n = 130) were treated with estradiol-17β for 3 days and then given an IPRD (Sincrogest, Ouro Fino, Brazil) 24 h after the last estradiol injection, and (4) RE-IPRD: acyclic mares (n = 32) were synchronized as in the IPRD group but not used for ET, then 8 to 15 days later, the mares were resynchronized with estradiol-17β for 3 days and IPRD for 4–8 days. 

All mares received a Day-8 in vivo-produced embryo (graded as 1 or 2, as previously described [28]), between 4 and 8 days after ovulation (Cyclic group) or progesterone treatments (acyclic groups), using a standard nonsurgical, transcervical technique. The protocols used for the synchronization of embryo recipient mares are highlighted in Figure 3. Mares in IPRD and RE-IPRD had the devices removed immediately before ET, and then a new IPRD was replaced after the ET. Immediately before removal of the IPRD, the vaginal inflammation and secretions were evaluated and scored as described in Experiment 1. Mares with severe mucopurulent discharge were not used for ET.

Pregnancy diagnosis (PD) was performed 5, 30, and 60 days after the ET by B-mode transrectal ultrasonography coupled with a 5 MHz transducer (Sono Scape A5V, Domed, São Paulo, Brazil). Once pregnancy was first confirmed, mares in acyclic groups (LAP4, IPRD, RE-IPRD) received weekly injections of LAP4 (1.5 g, IM) until 120 days of pregnancy. Mares in IPRD and RE-IPRD groups had the device removed three days after the first PD (Figure 3). 

#### Breeding Management of Embryo Donor Mares, Embryo Flushing, and Transfer

Eighty-six mares (Quarter Horse, Mangalarga Marchador, Mangalarga Paulista, Brazilian de Hipismo, and Arabian) with ages varying between four and twenty-one years old served as embryo donors. The mares were kept in private farms in the state of São Paulo, Brazil. Embryo donor mares were evaluated via transrectal palpation and ultrasonography, every other day until a pre-ovulatory follicle was detected (≥35 mm of diameter) when they had the ovulation induced with a single dose of histrelin acetate (250 μg, i.m., Strelin^®^, Botupharma, Botucatu, Brazil) or hCG (1500 IU, i.v., Chorulon^®^, MSD Saúde Animal, São Paulo, Brazil). Donor mares were artificially inseminated (AI), with at least 1 billion progressively motile sperm, 24 h after the induction of ovulation. Mares were treated post-AI for intrauterine fluid accumulation in standard fashion with oxytocin and uterine lavage. Embryo flushing was carried out eight days after ovulation with Lactate Ringer’s solution [29]. The recovered uterine fluid was drained into a filter, and embryos were searched, located, separated, and graded [28]. Only embryos of grades 1 and 2 were transferred.

Following collection, the embryos were packed in 5 mL vials containing embryo holding medium (Botuembryo^®^, Botupharma, Botucatu, Brazil) and shipped in a commercial container (Botuflex^®^, Botupharma, Botucatu, Brazil) at room temperature. The embryos were transported for no more than 6 h to the embryo recipient farm. After arrival on the farm, each embryo was examined under a stereomicroscope before being loaded into a 0.5 mL French straw. The French straw was then placed in an equine embryo transfer French gun (IMV Technologies, Campinas, Brazil) and covered with a plastic protector for ET.

### 2.3. Statistical Analyses

Data analyses were performed with GraphPad Prism 8.0.1. (GraphPad Software, San Diego, CA, USA). The Gaussian distribution of serum progesterone concentration was evaluated using the Kolmogorov–Smirnov normality test. Repeated measures ANOVA was followed by Tukey′s post hoc test to compare progesterone concentrations. Endometrial edema and uterine tone were assessed by the Kruskal–Wallis test followed by Dunn’s test. Conception rates were evaluated by the logistic regression model by considering pregnancy rate and loss as the dependent variables and breeding season and treatment groups as explanatory variables. Significance was set at *p* < 0.05 for all tests. All data are presented as a mean ± SD. The degree of linear correlation between the vaginal inflammation score and the day of the recipient was used for ET, and pregnancy rates and pregnancy losses were tested with Pearson’s coefficient of correlation. Strong coefficient of correlation was defined as r > 0.68, moderate 0.36 ≤ r ≤ 0.68, and weak correlation when r < 0.36 [30].

## 3. Results

### 3.1. Experiment 1

Mares had increased endometrial edema scores after estradiol-17β treatment, which were reduced two days after IPRD insertion (*p* < 0.05; Figure 4). The uterine tone increased after estradiol-17β treatment (*p* < 0.05) but not with IPRD (*p* > 0.05). Progesterone was rapidly absorbed two hours after IPRD insertion (Figure 5A). Insertion of IPRD resulted in a progesterone concentration deemed satisfactory to maintain pregnancy up to day 9 post-insertion (7.7 ± 2.3 ng/mL) (Figure 6). Progesterone concentrations declined rapidly two hours after removal of the IPRD (Figure 5B) and returned to baseline (<1 ng/mL) 24 h later (Figure 5 and Figure 6).

Three days after removal of IPRD, mares were treated again with estradiol-17β, and changes in endometrial edema were observed one day after treatment (*p* < 0.05; Figure 4A), but not in uterine tone (*p* > 0.05; Appendix A). Similarly, as observed in the first synchronization, IPRD insertion reduced endometrial edema (*p* < 0.05; Figure 4B) and increased serum progesterone concentration (*p* < 0.05; Figure 6B). One day post-removal of IPRD, serum progesterone returned to baseline (Figure 6B).

Seven mares (58%, 7/12) had no vaginal discharge after IPRD in the first synchronization. The remaining mares (42%, 5/12) presented a mild vaginal inflammation. After removing IPRD in the resynchronization, six mares (50%, 6/12) did not present any sign of vaginal inflammation, while the remaining mares (50%, 6/12) had a mild inflammatory reaction to the IPRD.

### 3.2. Experiment 2

One mare in the IPRD group had severe vaginal discharge detected at D8 and was not used for ET. In nineteen (15%, 19/130) mares, there was no vaginal discharge after IPRD removal for ET. Ninety mares (69%, 90/130) had a mild score for vaginitis, and twenty-one (16%, 21/130) received a moderate score. There was a moderate correlation (r = 0.47) between the vaginal discharge score and the day that the embryo recipient mare was used for ET (Table 1). Duration of IPRD in mares had a positive correlation with vaginal discharge scores, and mares that had the IPRD for longer periods (D7 and D8) had more severe vaginal inflammation than mares with the IPRD for shorter periods (D4 and D5). Six mares in the RE-IPRD group had no vaginal discharge (22%, 7/32), ten had mild (37%, 12/32), and twelve moderate (41%, 13/32) vaginal discharge. Resynchronization did not increase the chance for a more severe vaginal discharge score (*p* > 0.05). Pregnancy rates (r = −0.03) and losses (r = 0.06) were not correlated with the day of synchronization of the embryo recipient mare (Table 1). 

There was no effect of the breeding season on pregnancy rates or losses (*p* > 0.05). Pregnancy rates or losses were similar across groups (*p* > 0.05) (Figure 7). In addition, there was no effect of the day of synchronization on pregnancy rates or losses for mares having embryos transferred on days 4, 5, 6, 7, or 8 post-ovulation (*p* > 0.05). Resynchronized mares using IPRD had similar pregnancy rates and losses than mares used in the first synchronization protocol (*p* > 0.05, Figure 7).

## 4. Discussion

This study was set-forth to test IPRD as an alternative to LAP4 for synchronization of anestrus embryo recipient mares. In experiment 1, we evaluated the dynamics of progesterone concentration after insertion and removal of an IPRD in acyclic mares, and then before and after removal of a second IPRD. Our findings demonstrated that progesterone concentrations rapidly increase and decrease after insertion and removal of the IPRD device used herein. Previously, other authors have shown that CIDR, another type of IPRD, increased progesterone concentrations in plasma of mares [22,31,32,33,34]. Experiment 2, conducted in a large embryo transfer operation, demonstrated that the IPRD synchronization protocol used herein was equally effective in establishing satisfactory pregnancy rates in acyclic embryo recipient mares and did not result in greater pregnancy losses when compared to cyclic mares or those synchronized with LAP4. In addition, resynchronization of mares in the RE-IPRD group did not affect pregnancy rates and losses or scores for vaginitis. A recent small study conducted (n = 24) after the conclusion of the present study reported similar pregnancy rates and losses to our study using the device used herein [33].

Both synchronization protocols used herein mimicked the physiology of the estrous cycle in mares. Estradiol was administered for three days to emulate the estrogen being produced by the dominant follicle, and then a dose of LAP4 or insertion of the IPDR meant to emulate an increase in progesterone following ovulation [8]. Previous studies that used a similar approach with different progestins resulted in satisfactory pregnancy rates, similar to ours [6,7,9,10,11,35]. 

Progesterone is the main hormone responsible for the maintenance of pregnancy up to 100 days of gestation [36,37,38]. Administration of exogenous progestins is known to efficiently maintain pregnancy in acyclic, transitional, and ovariectomized mares, or mares suffering from luteal insufficiency [4,6,7,9,10,12,39,40,41,42]. The IPRD used in the present study rapidly increased progesterone concentrations well above the minimal threshold (2–4 ng/mL) deemed necessary to maintain pregnancy in mares during the first trimester [8]. Other studies using different IPRD in horses demonstrated that their devices were able to maintain serum progesterone (>2.5 ng/mL) for 12 days [22,32]. However, since embryo recipient mares are used for ET up to 9 days post-ovulation or progesterone treatment [5,6,7,43,44], we elected to keep the IPRD for 9 days. Additionally, while the present study did not keep IPRD for longer than 9 days, the progesterone concentration of 8 ng/mL upon removal suggests that the device would certainly be able to keep progesterone concentrations above the referenced value. 

Many practitioners refrain from using IPRD in mares due to the risk of vaginitis [20,22]. In experiment 2, the occurrence of vaginitis was severe in 1 out of 163 embryo recipient mares (IPRD + RE-IPRD groups). Eighty-four percent of mares had none or mild vaginal discharge scores, and this low incidence could be partially attributed to a good hygienic technique while inserting the device and to the use of oxytetracycline hydrochloride and hydrocortisone spray immediately before insertion of each IPRD. Although the single treatment with antibiotic spray into the IPRD goes against current recommendations of the World Health Organization, this therapy has been suggested by others to prevent/reduce the risk of IPRD-induced vaginitis in mares [27]. It is unknown if the corticoid suppresses local mucosal inflammation, or if the antibiotic suppresses the bacterial growth. While this remains to be determined, likely, the corticoid might be responsible for the low incidence of vaginitis obtained herein. Most importantly, there was no association between vaginal discharge score and pregnancy rates or losses in mares in the present study, as it has been described using IPRD to induce cyclicity in transitional mares [45]. Some clients could have been disturbed by the presence of vaginitis in their pet mare; however, others would probably accept the risk and would opt-in for using an IPRD, and most embryo recipient mares are not pets. While vaginitis induced by IPRD can be utterly impressive in some cases, the condition is self-resolving and requires no treatment [27,45]. 

Immediately before ET, the first IPRD was removed, and it was replaced after the ET. This was performed to allow satisfactory progesterone concentrations immediately after the transfer until three days after the PD. Replacing the IPRD before transfer likely minimizes the chances of contamination with vaginal secretions, as when secretion was present, it was scooped out with the first IPRD. The second IPRD was removed 3 days after the first PD. The authors elected not to remove immediately after the first PD because of the rapid decline in progesterone starting 2 h after removal of the IPRD shown in experiment 1; thus, mares received an injection of LAP4 at the first PD and then three days later the IPRD was removed. 

The hypothesis that mares synchronized with IPRD would respond to a resynchronization in a short period after IPRD removal was also confirmed in the present study. One day after IPRD removal, progesterone concentrations returned to baseline. Therefore, mares were resynchronized three days after the onset of synchronization with IPRD, and all mares had endometrial edema after estrogen treatment. Previous studies, treating seasonal transitional mares with IPRD to hasten cyclicity, demonstrated that mares display endometrial edema and estrus after IPRD removal [19,20,21,22,46]. In clinical practice, it is well-known that mares treated with long-acting altrenogest or LAP4 do not respond to estrogen treatment at the onset of a new synchronization protocol. Although long-acting altrenogest appears to be metabolized faster than LAP4, both hormones have a delayed metabolization after injection [7]. In addition, the presence of uterine edema is important for satisfactory pregnancy rates in embryo recipient mares [47], since estrogen is responsible for activating progesterone receptors in the endometrium [48,49,50,51]. In another study by our group, mares with lower endometrial edema scores during estrogen treatment had lower pregnancy rates compared with mares that had higher endometrial edema scores [7]. This finding supports the importance of endometrial edema prior to progesterone treatment for satisfactory establishment of pregnancy in embryo recipient mares [47]. Mares resynchronized between eight and fifteen days after the first synchronization herein had pregnancy rates similar to cyclic and acyclic mares treated with LAP4, and those synchronized with IPRD just once.

## 5. Conclusions

In conclusion, IPRD used herein resulted in a rapid increase in progesterone and a sharp decline upon its insertion and removal, respectively. Mares responded with endometrial edema after estradiol-17β administration after the onset of synchronization using IPRD as a progesterone source. Progesterone concentration also increased in the resynchronization of mares. Although vaginal inflammation was observed after IPRD in mares, it did not affect pregnancy rates or losses in embryo recipient mares after ET. Finally, this device can be used as a safe and compatible alternative to LAP4 to synchronize acyclic embryo recipient mares and it enables resynchronization of embryo recipient mares in a short period. 

## Figures and Tables

**Figure 1 vetsci-08-00190-f001:**
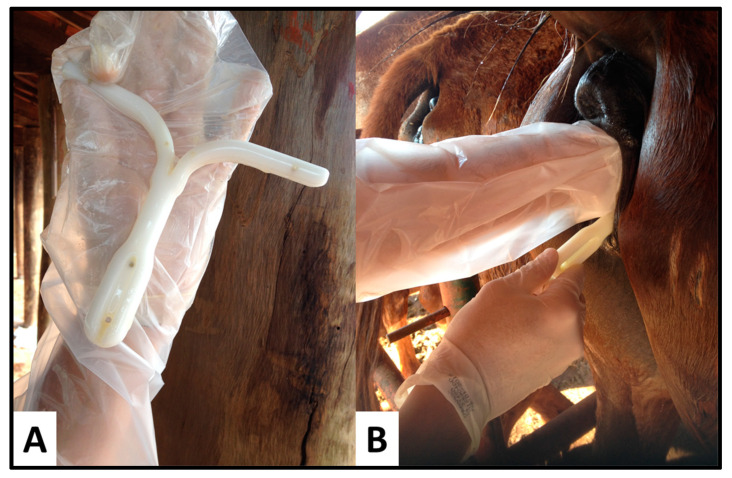
Placement of an intravaginal progesterone device (IPRD) (Sincrogest, Ourofino) in an acyclic mare. (**A**) The IPRD in the operator’s hand immediately after being sprayed with oxytetracycline and hydrocortisone. (**B**) IPRD while being inserted into the vagina of an embryo recipient mare.

**Figure 2 vetsci-08-00190-f002:**
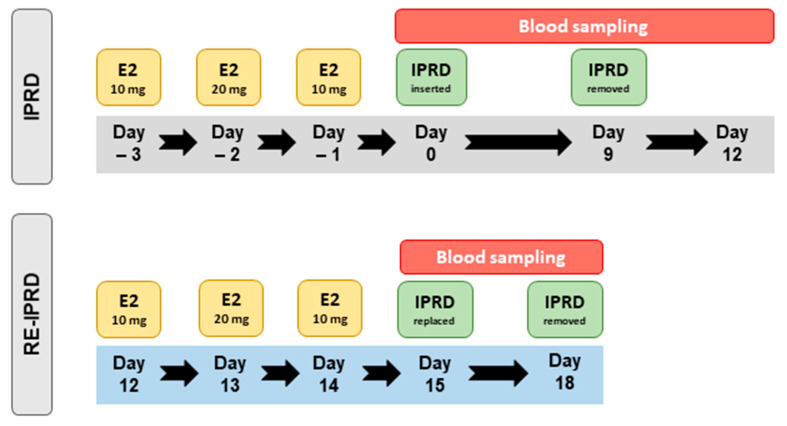
Timeline for blood sampling and hormone treatments administered to mares synchronized (IPRD) and resynchronized (RE-IPRD) with an intravaginal progesterone-releasing device (IPRD). Estradiol-17β (E2) (17 βeta^®^, 10 mg, IM); IPRD (1 g of natural progesterone). Day 0 is equivalent to the day of ovulation. Initially, acyclic mares were treated with estradiol-17β for 3 days and then given an IPRD (Day 0). The IPRD was kept for nine days and then removed (Day 9). Mares were resynchronized three days after removal of the IPRD (i.e., Day 12, after the first synchronization). Estradiol-17β was given for three days (Day 12–Day 14), and then had a new IPRD inserted (Day 15) and removed three days later (Day 18).

**Figure 3 vetsci-08-00190-f003:**
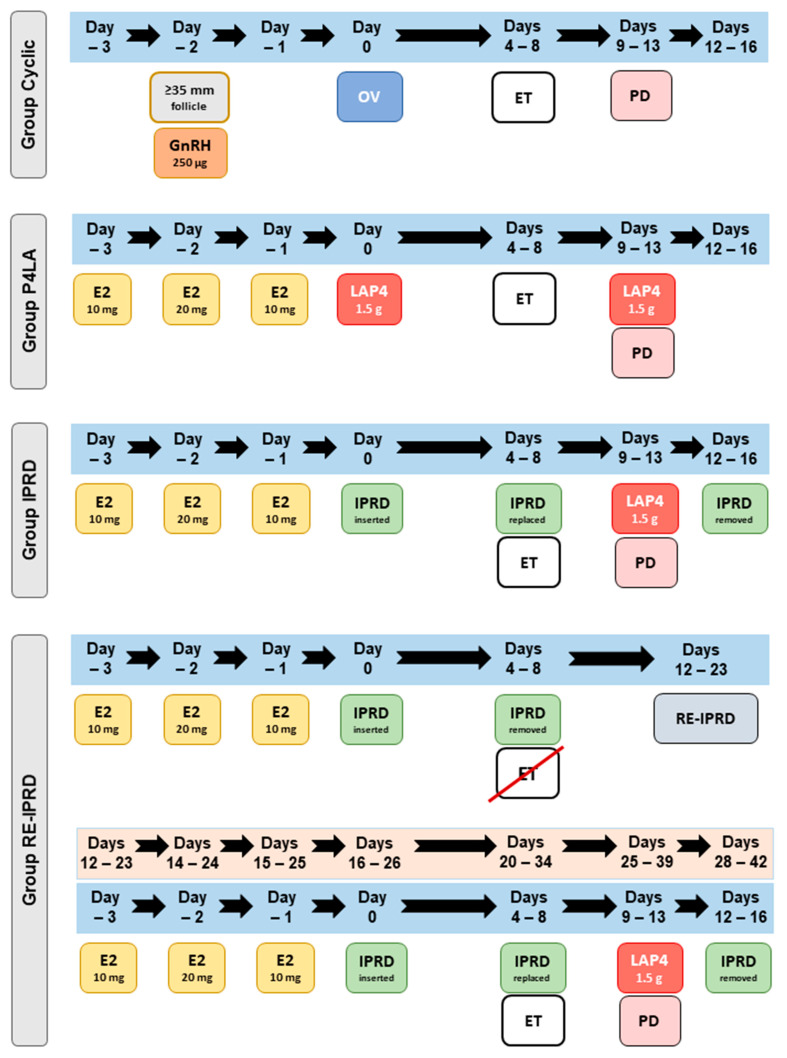
Timeline for hormone treatments administered to synchronize and resynchronize embryo recipient mares for embryo transfer. Cyclic: cyclic mares having ovulation confirmed after induction of ovulation (follicle ≥35 mm) with histrelin acetate (250 µg); LAP4: acyclic mares treated with estradiol-17β for 3 days (E2-3d) and then administered a single dose of long-action progesterone (LAP4, 1.5 g); IPRD: acyclic mares treated with E2-3d and then given an IPRD containing 1 g of natural progesterone; RE-IPRD: acyclic mares synchronized as in the IPRD group but not used for embryo transfer, then 8 to 15 days later (i.e., D12–D23 after the first synchronization), the mares were resynchronized with E2-3d and IPRD for 4–8 days. Embryo transfers (ET) were performed between 4 and 8 days after ovulation or progesterone treatment. Pregnancy diagnosis (PD) was performed five days after ET (Days 9–13). Pregnant embryo recipient mares from groups LAP4, IPRD, and RE-IPRD received weekly injections of LAP4 at PD. Pregnant mares in Groups IPRD and RE-IPRD had the device removed three days after the first PD.

**Figure 4 vetsci-08-00190-f004:**
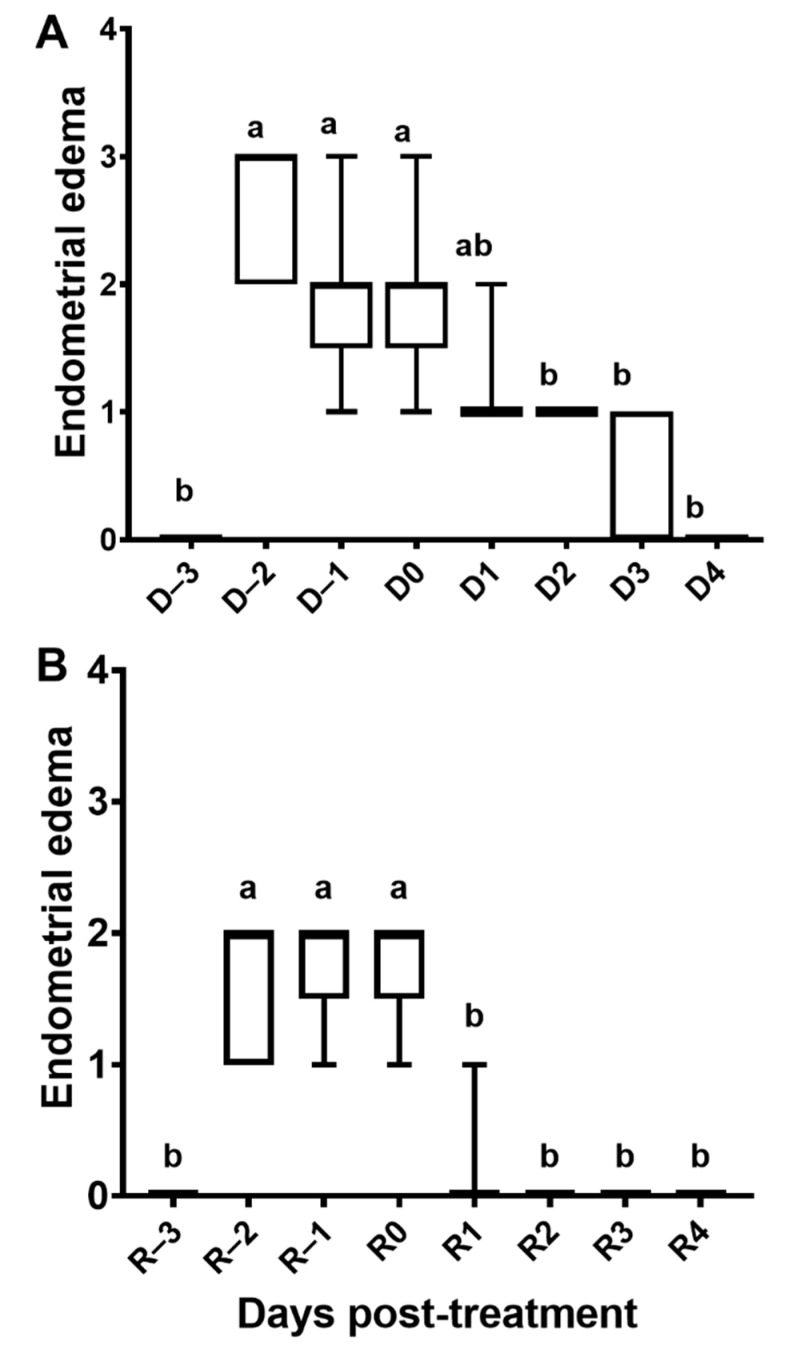
Endometrial edema scores of acyclic mares (n = 12) synchronized with estradiol and intravaginal progesterone release device (IPRD). (**A**) Mares treated with estradiol-17β for 3 days (D–3 to D–1) and then given an IPRD (D0). The IPRD was kept in place for nine days and then removed (D9). Mares did not have endometrial edema after day 3 (D3). (**B**) Mares were resynchronized three days post-removal of IPRD after the first synchronization. Mares were again treated with estradiol-17β for three days (R–3 to R–1) and then had a new IPRD inserted (R0). The IPRD was removed three days later (R3). Score 0 represents no edema, and score 4 represents exacerbated endometrial edema. D, day after treatment with IPRD; R, day after resynchronization with IPRD. Different super-scripts (^a,b^) denote effect of days (*p* < 0.05).

**Figure 5 vetsci-08-00190-f005:**
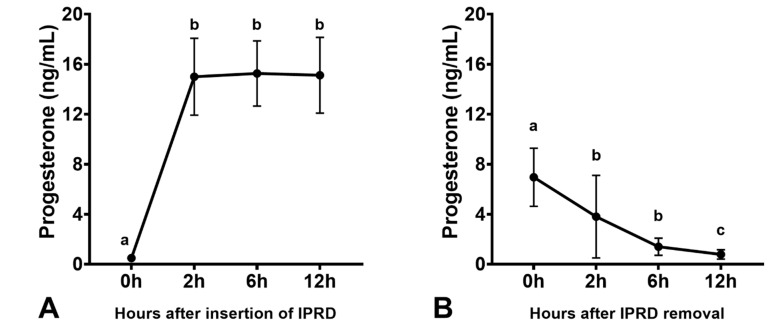
Serum progesterone concentration at 2, 6, and 12 h after insertion (**A**), and removal (**B**) of intravaginal progesterone-releasing device (IPRD) in acyclic mares (n = 12). Acyclic mares were treated with estradiol-17β for 3 days and then given an IPRD (A, 0 h). The IPRD was kept for nine days and was then removed (B, 0 h). Different superscripts (^a,b,c^) denote effect of time (*p* < 0.05).

**Figure 6 vetsci-08-00190-f006:**
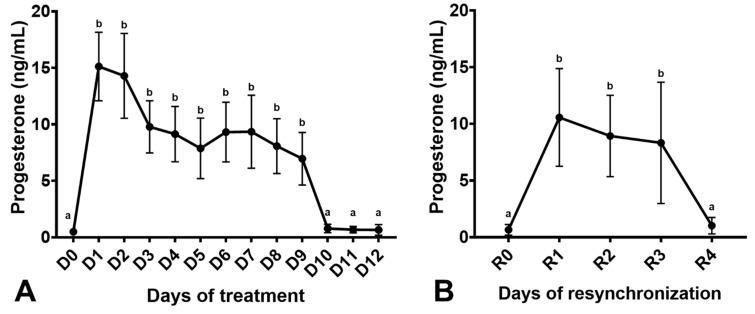
Serum progesterone concentrations in acyclic mares (n = 12) treated with intravaginal progesterone-releasing device (IPRD). (**A**) Mares were treated with estradiol-17β for 3 days and then given an IPRD (D0). The IPRD was removed nine days later (D9). (**B**) Mares were resynchronized three days after removal of IPRD after the first synchronization. Mares were again treated with estradiol-17β for 3 days and then had a new IPRD inserted (R0). The IPRD was kept for three days (R3). D, day after treatment with IPRD; R, day after resynchronization with IPRD. Different superscripts (^a,b^) denote effect of days (*p* < 0.05).

**Figure 7 vetsci-08-00190-f007:**
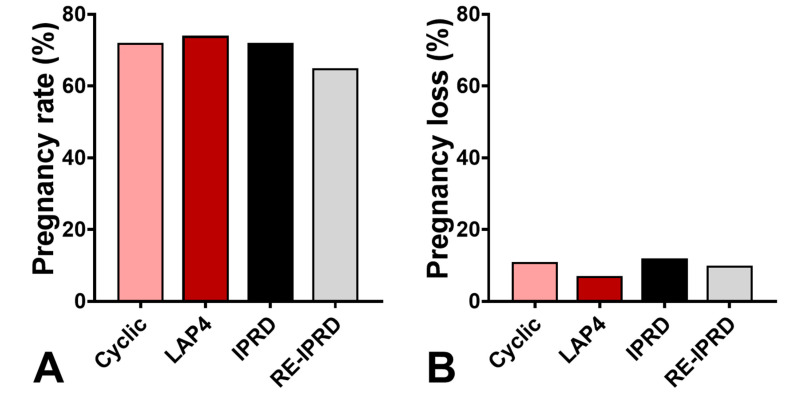
Pregnancy rates (**A**) and pregnancy losses (**B**) up to 60 days of cyclic and acyclic mares synchronized with long-acting progesterone (LAP4) or intravaginal progesterone-releasing device for the first time (IPRD, n = 130) or resynchronized with IPRD (n = 32) between 8 and 15 days after the first synchronization (RE-IPRD). Cyclic (n = 75): mares having ovulation confirmed after induction of ovulation (follicle ≥ 35 mm) with histrelin acetate (250 µg); LAP4 (n = 92): acyclic mares treated with estradiol-17β for 3 days and then administered a single dose of long-action progesterone (LAP4, 1.5 g); IPRD (n = 130): acyclic mares treated with estradiol-17β for 3 days and then given an IPRD containing 1 g of natural progesterone; RE-IPRD: mares not used after 8 days after insertion of an IPRD in the first synchronization were treated 8–15 days later with estradiol-17β for 3 days and then given a new IPRD. Embryo transfer was performed 4–8 days after ovulation or progesterone treatment and pregnancy diagnosis was performed 5 days later. Once pregnancy was confirmed, mares in the 3 acyclic groups received weekly injections of LAP4 (1.5 g) until 120 days of pregnancy. Mares in IPRD and RE-IPRD groups had the device removed three days after the first PD.

**Table 1 vetsci-08-00190-t001:** Vaginal discharge, pregnancy rates, and losses of embryo recipient mares synchronized with an intravaginal progesterone-releasing device at D4 to D8.

Day of Synchronization	D4	D5	D6	D7	D8	Total
**Categories**						
**Absent**	11	6	2	0	0	19
**Mild**	24	19	27	11	9	90
**Moderate**	2	0	6	8	5	21
**Severe**	0	0	0	0	1	1
**Total**	37	25	35	19	15	131
**Pregnancy rates**	73% (27/37)	76% (19/25)	69% (24/35)	68% (13/19)	71% (10/14)	72% (93/130)
**Pregnancy losses**	11% (3/27)	11% (2/19)	8% (2/24)	15% (2/13)	20% (2/10)	12% (11/93)

## Data Availability

The original contributions presented in the study are included in the article/Appendix A, and further inquiries can be directed to the corresponding author/s.

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
