# Peer review of "Use of Intravaginal Progesterone-Releasing Device Results in Similar Pregnancy Rates and Losses to Long-Acting Progesterone to Synchronize Acyclic Embryo Recipient Mares"

_vetsci, 2021, doi:10.3390/vetsci8090190_

Round 1
Reviewer 1 Report
General comments: The study is very well designed and executed, and the results have important clinical implications in equine embryo transfer. However, the manuscript requires some changes before it can be recommended for publication. Specific comments for improvement are provided below:
Specific comments:
Abstract:
Lines 15-17: This is a run on sentence and needs to rewritten. I also suggest being consistent with either "embryo recipient" or "embryo surrogate" throughout the manuscript.
Line 23: Delete "as above"
Line 49: I suggest "achieved" instead of "reached"
Lines 50-52: This sentence reads awkward. I suggest rephrasing it or splitting it into two shorter and clearer sentences.
Lines 54-55: "as long a progestins administered"...the authors probably meant "as long as progestins are administered"
Materials and methods:
Lines 86-87: "absence of endometrial edema, corpus luteum, follicles < 20 mm of diameter and serum progesterone concentration <1 ng/mL"....Since the authors are beginning with "absence of", shouldn't it be "follicles > 20 mm" and "progesterone concentration >1 ng/mL" instead?
Lines 112-113: Why is there a discrepancy between the time points (0h, 2h, 6h and 12h) listed here and those shown in Figure 5 (0h, 2h, 6h and 24h)?
Also, the statement is unclear and must be rewritten to clearly indicate how many daily samples were collected during synchronization and resynchronization.
Line 116: "blood sampling" instead of "blood samplings"
Lines 123: "Blood samples were collected" instead of "Serum samples were collected"
Line 133: "a private"
Line 135: "aging from 3 to 12 years-old" reads odd.....maybe "aged 3 to 12 years" or simply "3- to 12-year-old"
Line 168: "Embryo transfers were"
Line 186: "5 mL of embryo medium"
Results:
The results are well written and presented.
Discussion:
Line 302: "CIDR another type of IPRD".....insert commas after CIDR and IPRD
Line 317: "is" instead of "are"
Lines 318-319: "acyclic, transitional, ovariectomized or mares suffering from luteal insufficiency" reads awkward. I suggest "acyclic, transitional, and ovariectomized mares, or mares suffering from luteal insufficiency"
Line 329: "in" instead of "on"
Line 339-340: Is this anecdotal or do you have a reference you can cite for this statement?
Line 343: Replace "minimizing" with "minimizes"
Line 356: "In clinical practice is well know"....replace with "In clinical practice, it is well known"
Line 361: "e progesterone"....Delete "e"
Line 376: Replace "safely" with "safe"
Author Response
Response to reviewer 1:
General comments: The study is very well designed and executed, and the results have important clinical implications in equine embryo transfer. However, the manuscript requires some changes before it can be recommended for publication. Specific comments for improvement are provided below:
Specific comments:
Abstract:
Query#1. Lines 15-17: This is a run on sentence and needs to rewritten. I also suggest being consistent with either "embryo recipient" or "embryo surrogate" throughout the manuscript.
Reply: The sentence was revised as suggested and the terminology “embryo-recipient” was used throughout the manuscript.
Query#2. Line 23: Delete "as above"
Reply: Deleted as suggested.
Query#3. Line 49: I suggest "achieved" instead of "reached"
Reply: Edited as suggested.
Query#4. Lines 50-52: This sentence reads awkward. I suggest rephrasing it or splitting it into two shorter and clearer sentences.
Reply: It was rephrased as suggested.
Query#5. Lines 54-55: "as long a progestins administered"...the authors probably meant "as long as progestins are administered"
Reply: This typo was corrected.
Materials and methods:
Query#6. Lines 86-87: "absence of endometrial edema, corpus luteum, follicles < 20 mm of diameter and serum progesterone concentration <1 ng/mL"....Since the authors are beginning with "absence of", shouldn't it be "follicles > 20 mm" and "progesterone concentration >1 ng/mL" instead?
Reply: Edited as suggested.
Query#7. Lines 112-113: Why is there a discrepancy between the time points (0h, 2h, 6h and 12h) listed here and those shown in Figure 5 (0h, 2h, 6h and 24h)?
Reply: This typo was corrected; thanks for such a careful review.
Query#8. Also, the statement is unclear and must be rewritten to clearly indicate how many daily samples were collected during synchronization and resynchronization.
Reply: The statement was revised as suggested.
Query#9. Line 116: "blood sampling" instead of "blood samplings"
Reply: The typo was corrected.
Query#10. Lines 123: "Blood samples were collected" instead of "Serum samples were collected"
Reply: Edited as suggested.
Query#11. Line 133: "a private"
Reply: This typo was fixed .
Query#12. Line 135: "aging from 3 to 12 years-old" reads odd.....maybe "aged 3 to 12 years" or simply "3- to 12-year-old"
Reply: Edited as suggested.
Query#13. Line 168: "Embryo transfers were"
Reply: The typo was corrected.
Query#14. Line 186: "5 mL of embryo medium"
Reply: Edited as suggested.
Results:
The results are well written and presented.
Discussion:
Query#15. Line 302: "CIDR another type of IPRD".....insert commas after CIDR and IPRD
Reply: Edited as suggested.
Query#16. Line 317: "is" instead of "are"
Reply: This typo was corrected.
Query#17. Lines 318-319: "acyclic, transitional, ovariectomized or mares suffering from luteal insufficiency" reads awkward. I suggest "acyclic, transitional, and ovariectomized mares, or mares suffering from luteal insufficiency"
Reply: Edited as suggested.
Query#18. Line 329: "in" instead of "on"
Reply: The typo was corrected.
Query#19. Line 339-340: Is this anecdotal or do you have a reference you can cite for this statement?
Reply: Thanks for the question, a reference was added to support this statement.
Query#20. Line 343: Replace "minimizing" with "minimizes"
Reply: Changed as suggested.
Query#21. Line 356: "In clinical practice is well know"....replace with "In clinical practice, it is well known"
Reply: Edited as suggested.
Query#22. Line 361: "e progesterone"....Delete "e"
Reply: It was deleted.
Query#23. Line 376: Replace "safely" with "safe"
Reply: It was replaced.
Reviewer 2 Report
Having analyzed the paper and verified that it is a comparative technical application test between two systems known for a long time of controlled release of progesterone, this should be clearly already in the title. This would greatly help readers to better understand the dynamics of blood progesterone, determined with correct analysis techniques.
The paper does not highlight anything specific about the release matrices, both as an injectable and as an intravaginal device.
The single treatment with intravaginal antibiotic at the time of insertion of the device is anachronistic and against the common supranational rules (WHO). The advisability of using non-antibiotic treatments as a single deep intravaginal irrigation before implantation of the controlled progesterone release system should be discussed and perhaps also in the conclusions. All this has the aim of reducing the risk of antibiotic resistance, a phenomenon that is underestimated by many but really present in mares. Modulation of the vaginal microbiota with single antibiotic treatment can lead to the onset of vaginitis which could affect gestation.
As my consolidated scientific vision as a Clinical and applicative Veterinary Physiologist with a specific address in Animal Reproduction Physiology, I believe that the result of technical application tests and innovative scientific research in the field of animal reproduction is not the achievement of the establishment of gestation, but the birth of a new individual in good health with all the physiological conditions of being able to reproduce himself. If the authors were in possession of these data, the work would assume better technical characteristics and open to new reflections
Statistical analysis should be re-evaluated by an expert in the field so that a greater amount of useful information can be extrapolated to the clinical veterinarian.
Author Response
Replies to reviewer #2:
Query#1. Having analyzed the paper and verified that it is a comparative technical application test between two systems known for a long time of controlled release of progesterone, this should be clearly already in the title. This would greatly help readers to better understand the dynamics of blood progesterone, determined with correct analysis techniques.
Reply: We are grateful for the comments; however, it is unclear to us how we could improve the title. The manuscript aims to introduce a new protocol for synchronization of embryo recipient mares.
Query#2. The paper does not highlight anything specific about the release matrices, both as an injectable and as an intravaginal device.
Reply: Sentences were added in the introduction, and materials and methods to about each treatment to address this suggestion.
Query#3. The single treatment with intravaginal antibiotic at the time of insertion of the device is anachronistic and against the common supranational rules (WHO). The advisability of using non-antibiotic treatments as a single deep intravaginal irrigation before implantation of the controlled progesterone release system should be discussed and perhaps also in the conclusions. All this has the aim of reducing the risk of antibiotic resistance, a phenomenon that is underestimated by many but really present in mares. Modulation of the vaginal microbiota with single antibiotic treatment can lead to the onset of vaginitis which could affect gestation.
Reply: We have included a note in the discussion as suggested.
Query #4. As my consolidated scientific vision as a Clinical and applicative Veterinary Physiologist with a specific address in Animal Reproduction Physiology, I believe that the result of technical application tests and innovative scientific research in the field of animal reproduction is not the achievement of the establishment of gestation, but the birth of a new individual in good health with all the physiological conditions of being able to reproduce himself. If the authors were in possession of these data, the work would assume better technical characteristics and open to new reflections.
Reply: This is an interesting comment, but the authors are not in possession of such a data set. Recipients left the farm after the last pregnancy diagnosis and then were subjected to the management of the owner leasing or purchasing the recipient.
Query # 5. Statistical analysis should be re-evaluated by an expert in the field so that a greater amount of useful information can be extrapolated to the clinical veterinarian.
Reply: Statistical analyses have been evaluated by an expert, and all data associated with the study have been described.
Reviewer 3 Report
Abstract
L15 was this aim achieved? Perhaps a more positive opening sentence would be “This study assessed uterine features…”
L16 change to “….pregnancy rates and embryo losses in surrogate…”
L17 please clarify that the synchronized surrogates were compared to naturally cycling recipients. The use of the two terms for embryo recipients is confusing.
L18 acyclic mares were not mentioned in the first sentence, were the synchronized mares acyclic?
L20 when did the 11 consecutive days of blood collection begin? When the IRPD was inserted or removed?
L22 when in the cycle was histrelin administered?
L23 when was LAP4 administered
L23 when was the IRPD inserted for this group?
L24 were the Day-8 embryos in vitro produced?
L25 4-8 days after P4 treatments -
L27 how long was the IPRD left in place in group 3 after ET?
L28 Pregnancy checks were at D5, D30 and D90 after transfer – when was pregnancy confirmed and the LAP4 treatments begun?
L33 what happened in Experiment 1?
L37 if there were no differences in pregnancy rate or loss, the use of the IRPD
was superfluous and involved more animal manipulation
The descriptions of the experimental groups is quite confusing and, as written, it appears that there are 4 separate groups Exp 1, Exp 2.1, 2.2, 2.3. There is no mention of resynchronization in the abstract.
Introduction
Be consistent: using either ‘surrogate mares’ or ‘recipient mares’, but not both
L49 Does success depend on the developmental stage of the embryo? Ovulation 4-9 days before ET seems a wide range of uterine stages.
L54 ‘as long as progestins are administered…’
L72 there is no indication in the abstract that resynchronization of mares was an objective in this study
Materials and Methods
Was there animal welfare oversight for this project?
L89 a very brief evaluation of the body condition score will inform readers of the meaning of ±7
- what was the age range of the mares in Experiment 1?
L109 what was recorded in addition to edema scores?
L112 briefly describe the edema scores
L113 was this blood collection scheme followed for both IPRD insertions?
Figure 1. The first photo is informative. The second is not and should be omitted.
Figure 2. This is most helpful in understanding a very confusing experimental design.
L129 respectively?
L134 add longitude and latitude of this facility
- how was the management and diet of these mares different from Exp 1 mares?
L135 would the differences between ‘light breed’ and ‘crossbred’ mares influence results?
In Experiment 1 the mares were seasonally anestrus, in Experiment 2 they were anestrus during the breeding season. How might that affect the results?
L137 when was histrelin administered? How often was ultrasound performed?
L139 when was the LAP4 given in Group 2 mares?
L141-146 was the subgroup of mares from Group 3? This section is very confusing.
L151 was this procedure and these analyses performed on mares in Exp 1 as well?
L181 explain “were treated post-breeding for fluid…”. Use post-AI instead of post-breeding
L186 the embryos were packed in 5 mL what? vials? straws?
L199 Uterine tone must be described in the materials and methods with a scoring system
L203 vaginitis is not described using that term in the Materials and Methods
L249 poor uterine tone needs to be described – what does it feel like?
L250 ‘a well-toned uterus’ must be described
L253 these results are concerning – how does this influence your conclusions in the abstract that “IPRD can be used as a compatible alternative to LAP4”?
Figure 7 although not statistically significant, is there a concern about the higher rate of pregnancy loss in the IPRD group?
L355 “mares….are used to coming in estrus” – seems anthropomorphic
L361 remove ‘e’
The descriptions of experimental design are very confusing and need significant rewording.
Author Response
Reviewer #3:
Query#1. L15 was this aim achieved? Perhaps a more positive opening sentence would be “This study assessed uterine features…”
Reply: This sentence was edited as suggested.
Query#2. L16 change to “….pregnancy rates and embryo losses in surrogate…”
Reply: We appreciate the suggestion; we have changed through “embryo surrogate” for “embryo recipient” throughout as suggested by reviewer 1.
Query#3. L17 please clarify that the synchronized surrogates were compared to naturally cycling recipients. The use of the two terms for embryo recipients is confusing.
Reply: Yes, they were compared. This segment was edited as suggested to ensure clarity.
Query#4. L18 acyclic mares were not mentioned in the first sentence, were the synchronized mares acyclic?
Reply: Yes, they were acyclic, this note was included to ensure clarity.
Query#5. L20 when did the 11 consecutive days of blood collection begin? When the IRPD was inserted or removed?
Reply: This information was included to ensure clarity .
Query#6. L22 when in the cycle was histrelin administered?
Reply: When the mare had a periovulatory follicle, this note was included to ensure clarity.
Query#7. L23 when was LAP4 administered
Reply: A note was added to ensure clarity.
Query#8. L23 when was the IRPD inserted for this group?
Reply: A note was added to ensure clarity.
Query#9. L24 were the Day-8 embryos in vitro produced?
Reply: No, there were in vivo produced embryos. A note was added to ensure clarity.
Query#10. L25 4-8 days after P4 treatments -
Reply: After ovulation confirmed in the Cyclic group as stated in L22-23, and LA-P4 treatments in Group 2 and 3.
Query#11. L27 how long was the IPRD left in place in group 3 after ET?
Reply: The IPRD left for 3 more days after pregnancy diagnosis, as stated in L31-32.
Query#12. L28 Pregnancy checks were at D5, D30 and D90 after transfer – when was pregnancy confirmed and the LAP4 treatments begun?
Reply: Pregnancy diagnosis were performed 5, 30 and 60 days after ET. Once pregnancy was confirmed, mares in Groups 2, 3 and 4 received weekly injections of LAP4 (1.5 g) until 120 days of pregnancy. This was stated in L29-31.
Query#13. L33 what happened in Experiment 1?
Reply: A description was added as suggested to ensure clarity
Query#14. L37 if there were no differences in pregnancy rate or loss, the use of the IRPD was superfluous and involved more animal manipulation
Reply: The objective was not to increase fertility rates of embryo recipient mares. The goal of this study was to make an alternative protocol for LA progesterone, as it is stated in the introduction L61-78. The lack of difference is assuring to practitioners that the protocol described is a viable alternative to be used in embryo recipient farms. A mare synchronized with LAP4 takes 2-3 weeks to show edema in another unpublished study.
Query#15 The descriptions of the experimental groups is quite confusing and, as written, it appears that there are 4 separate groups Exp 1, Exp 2.1, 2.2, 2.3. There is no mention of resynchronization in the abstract.
Reply: The resynchronization was added in the abstract to ensure clarity.
Introduction
Query#16. Be consistent: using either ‘surrogate mares’ or ‘recipient mares’, but not both
Reply: We elected to use embryo recipient, thus, this was edited revised throughout the manuscript.
Query#17. L49 Does success depend on the developmental stage of the embryo? Ovulation 4-9 days before ET seems a wide range of uterine stages.
Reply: Thanks for the comment, this range is widely acceptable and used in horses. Embryo transfer is typically performed between 4 to 9 days post-ovulation of the embryo recipient or P4 treatments of embryo recipient mares.
- Greco, G.M.; Fioratti, E.G.; Segabinazzi, L.G.; Dell’Aqua, J.A.; Crespilho, A.M.; Castro-Chaves, M.M.B.; Alvarenga, M.A. Novel Long-Acting Progesterone Protocols Used to Successfully Synchronize Donor and Recipient Mares With Satisfactory Pregnancy and Pregnancy Loss Rates. J. Equine Vet. Sci. 2016, 39, doi:10.1016/j.jevs.2015.07.012.
- Filho, A.N.R.; Pessôa, M.A.; Gioso, M.M.; Alvarenga, M.A. Transfer of equine embryos into anovulatory recipients supplemented with short or long acting progesterone. Anim. Reprod. 2004, 1, 91–95.
- Hinrichs, K.; Sertich, P.L.; Kenney, R.M. Use of altrenogest to prepare ovariectomized mares as embryo transfer recipients. Theriogenology 1986, 26, 455–460, doi:10.1016/0093-691X(86)90037-3.
- McKinnon, A.O.; Squires, E.L.; Carnevale, E.M.; Hermenet, M.J. Ovariectomized steroid-treated mares as embryo transfer recipients and as a model to study the role of progestins in pregnancy maintenance. Theriogenology 1988, 29, 1055–1063, doi:10.1016/S0093-691X(88)80029-3.
- Oliveira Neto, I. V.; Canisso, I.F.; Segabinazzi, L.G.; Dell’Aqua, C.P.F.; Alvarenga, M.A.; Papa, F.O.; Dell’Aqua, J.A. Synchronization of cyclic and acyclic embryo recipient mares with donor mares. Anim. Reprod. Sci. 2018, 190, 1–9, doi:10.1016/j.anireprosci.2017.12.016.
- Botelho, J.H. V; Pessoa, G.O.; Rocha, L.G.P.; Yeste, M. Hormone supplementation protocol using estradiol benzoate and long-acting progesterone is efficient in maintaining pregnancy of anovulatory recipient mares during autumn transitional phase. Anim. Reprod. Sci. 2015, 153, 39–43.
- Allen, W.R.; Rowson, L.E. Surgical and non-surgical egg transfer in horses. J. Reprod. Fertil. Suppl. 1975, 525—530.
- Oliveira Neto, I.V.; Canisso, I.F.; Segabinazzi, L.G.; Dell’Aqua, C.P.F.; Alvarenga, M.A.; Papa, F.O.; Dell’Aqua, J.A. Synchronization of cyclic and acyclic embryo recipient mares with donor mares. Anim. Reprod. Sci. 2018, 190, doi:10.1016/j.anireprosci.2017.12.016.
- Wilsher, S.; Clutton-Brock, A.; Allen, W.R. Successful transfer of day 10 horse embryos: Influence of donor-recipient asynchrony on embryo development. Reproduction 2010, 139, 575–585, doi:10.1530/REP-09-0306.
- Jacob, J.C.F.; Haag, K.T.; Santos, G.O.; Oliveira, J.P.; Gastal, M.O.; Gastal, E.L. Effect of embryo age and recipient asynchrony on pregnancy rates in a commercial equine embryo transfer program. Theriogenology 2012, 77, 1159–1166, doi:10.1016/j.theriogenology.2011.10.022.
Query#18. L54 ‘as long as progestins are administered…’
Reply: Edited as suggested.
Query#19. L72 there is no indication in the abstract that resynchronization of mares was an objective in this study
Reply: This was added in the abstract to ensure clarity
Materials and Methods
Query#20. Was there animal welfare oversight for this project?
Reply: Yes, indeed, as stated in L80-81.
Query#21. L89 a very brief evaluation of the body condition score will inform readers of the meaning of ±7
Reply: The authors appreciate the reviewer’s suggestion. Since body score is a categorical variable, we included the median value.
Query#22. what was the age range of the mares in Experiment 1?
Reply: This was added.
Query#23. L109 what was recorded in addition to edema scores?
Reply: Uterine tone was also recorded. It was added in the M&M L119-121.
Query#24. L112 briefly describe the edema scores
Reply: It was added in the M&M L119-121
Query#25. L113 was this blood collection scheme followed for both IPRD insertions?
Reply: Yes, all mares were synchronized at the same time in Experiments 1.
Query#26. Figure 1. The first photo is informative. The second is not and should be omitted.
Reply: The authors appreciated the reviewer perspective. However, we feel that photo 1 would be less informative without photo 2. Therefore, we elected to keep both photos in figure 1.
Query#27. Figure 2. This is most helpful in understanding a very confusing experimental design.
Reply: Thanks for the comments.
Query#28. L129 respectively?
Reply: It was deleted.
Query#29. L134 add longitude and latitude of this facility
Reply: It was added as suggested(L144).
Query#30. - how was the management and diet of these mares different from Exp 1 mares?
Reply: The diet management of mares in Exp 2 was added as suggested.
Query#31. L135 would the differences between ‘light breed’ and ‘crossbred’ mares influence results?
Reply: We used crossbred of light breed mares.
Query#32. In Experiment 1 the mares were seasonally anestrus, in Experiment 2 they were anestrus during the breeding season. How might that affect the results?
Reply: We have added some context in the background hopefully it will be more clear to the reader, thanks for the suggestion.
Query#33. L137 when was histrelin administered? How often was ultrasound performed?
Reply: A single dose of histrelin was given when the mare had a periovulatory follicle.
Query#34. L139 when was the LAP4 given in Group 2 mares?
Reply: This is mentioned in the text and in Figure 3.
Query #35. L141-146 was the subgroup of mares from Group 3? This section is very confusing.
Reply: It was clarified in the sentence. The subgroup was called as Group 4.
Query#36. L151 was this procedure and these analyses performed on mares in Exp 1 as well?
Reply: Yes, this is included in the M&M section in Experiment 1 and in the results.
Query#37. L181 explain “were treated post-breeding for fluid…”. Use post-AI instead of post-breeding
Reply: Sentences were added to ensure clarity.
Query#38. L186 the embryos were packed in 5 mL what? vials? straws?
Reply: A note was included to ensure clarity.
Query#39. L199 Uterine tone must be described in the materials and methods with a scoring system
Reply: It was added in the M&M section as suggested.
Query #40. L203 vaginitis is not described using that term in the Materials and Methods
Reply: It was reviewed.
Query#41. L249 poor uterine tone needs to be described – what does it feel like?
Reply: Described as suggested. Feels floppy, and flaccid
Query #42. L250 ‘a well-toned uterus’ must be described
Reply: Described a suggested.
Query #43 L253 these results are concerning – how does this influence your conclusions in the abstract that “IPRD can be used as a compatible alternative to LAP4”?
Reply: The results shown that IPRD is a compatible alternative, as pregnancy rates and pregnancy losses up to 60 days are similar to cyclic mares or mares treated with LAP4. Therefore, this protocol can be used to synchronize embryo recipient mares with no negative impact on fertility results.
Query #44. Figure 7 although not statistically significant, is there a concern about the higher rate of pregnancy loss in the IPRD group?
Reply: No concern. This still within acceptable ranges for species when compared with other embryo transfer programs.
Query #45. L355 “mares….are used to coming in estrus” – seems anthropomorphic
Reply: This sentence was edited to ensure clarity .
Query #46. L361 remove ‘e’
Reply: It was removed.
Query #47. The descriptions of experimental design are very confusing and need significant rewording.
Reply: The experiment design was edited to ensure clarity. In addition, Figure 3 was created to help readers follow the design. In addition, subgroup 3 was labelled as group 4.
Reviewer 4 Report
General comments
This study mainly aimed to evaluate the use of an intravaginal progesterone releasing device as an alternative to the long-acting progesterone in ET surrogate mares. The study is very well designed and structured. The conclusions are supported by results. Only minor issues are detected.
Specific comments
L73: Please remove “i)”.
L129: Please remove “respectively” or add the inter-assay.
L159: Please add the model and frequency (MHz) used to perform the PD.
L233 (Fig. 6): What values are different?
L324: Be consistent in the text (progesterone instead of P4).
Author Response
Reviewer #4:
General comments
This study mainly aimed to evaluate the use of an intravaginal progesterone releasing device as an alternative to the long-acting progesterone in ET surrogate mares. The study is very well designed and structured. The conclusions are supported by results. Only minor issues are detected.
Specific comments
Query #1. L73: Please remove “i)”.
Reply: Removed as suggested.
Query #2. L129: Please remove “respectively” or add the inter-assay.
Reply: Edited as suggested.
Query #3. L159: Please add the model and frequency (MHz) used to perform the PD.
Reply: Added as suggested.
Query #4. L233 (Fig. 6): What values are different?
Reply: Figure 6 was modified to show the difference between P4 levels.
Query #5. L324: Be consistent in the text (progesterone instead of P4).
Reply: It was revised throughout as suggested.
Round 2
Reviewer 3 Report
The authors' responses to this review are commendable. The manuscript is greatly improved and is now acceptable for publication.